mathematical modelling/health and disease and epidemiology

COVID-19, non-pharmaceutical interventions, hospital setting, individual-based model

**Authors for correspondence:**
Hongyang Zhao
e-mail: hyzhao750@sina.com
Martial Ndeffo-Mbah
e-mail: mndeffo@cvm.tamu.edu

†These authors contributed equally to this study.

# SARS-CoV-2 transmission and control in a hospital setting: an individual-based modelling study

Qimin Huang[1,†], Anirban Mondal[1,†], Xiaobing Jiang[3,†], Mary Ann Horn[1], Fei Fan[3], Peng Fu[3], Xuan Wang[3], Hongyang Zhao[3], Martial Ndeffo-Mbah[4,5] and David Gurarie[1,2]

[1]Department of Mathematics, Applied Mathematics, and Statistics, and [2]Center for Global Health and Diseases, School of Medicine, Case Western Reserve University, Cleveland, OH 44106, USA
[3]Department of Neurosurgery, Union Hospital, Tongji Medical College, Huazhong University of Science and Technology, Wuhan 430022, People's Republic of China
[4]Department of Veterinary and Integrative Biosciences, College of Veterinary and Biomedical Sciences, and [5]School of Public Health, Texas A&M University, College Station, TX 77840, USA

QH, 0000-0001-6994-1739; AM, 0000-0002-4100-2366; HZ, 0000-0003-3684-6297; MN-M, 0000-0003-4158-7613; DG, 0000-0002-5314-7888

Development of strategies for mitigating the severity of COVID-19 is now a top public health priority. We sought to assess strategies for mitigating the COVID-19 outbreak in a hospital setting via the use of non-pharmaceutical interventions. We developed an individual-based model for COVID-19 transmission in a hospital setting. We calibrated the model using data of a COVID-19 outbreak in a hospital unit in Wuhan. The calibrated model was used to simulate different intervention scenarios and estimate the impact of different interventions on outbreak size and workday loss. The use of high-efficacy facial masks was shown to be able to reduce infection cases and workday loss by 80% (90% credible interval (CrI): 73.1–85.7%) and 87% (CrI: 80.0–92.5%), respectively. The use of social distancing alone, through reduced contacts between healthcare workers, had a marginal impact on the outbreak. Our results also indicated that a quarantine policy should be coupled with other interventions to achieve its effect. The effectiveness of all these interventions was shown to increase with their early implementation. Our analysis shows that a COVID-19 outbreak in a hospital's non-COVID-19 unit can be controlled or mitigated by the use of existing non-pharmaceutical measures.

# 1. Introduction

The world is in the midst of an unprecedented coronavirus outbreak caused by a novel virus recently named COVID-19 by the World Health Organization (WHO). Developing strategies for mitigating the severity of COVID-19 is now a top global health priority. The range of containment strategies employed in different countries and regions varies from shelter-in-place orders, the shutdown of public events, travel ban [1] and visitor quarantine, to intermediate steps that involve partial closures (e.g. schools [2], workplaces, sporting and cultural events) [3]. While such drastic steps can reduce infection spread, they exact a heavy toll on society and human well-being. At present, the only available means of containing COVID-19 spread is via the use of non-pharmaceutical interventions [4,5] such as social distancing, self-isolation [6], tracing and quarantine [6,7], wearing facial masks/personal protective equipment (PPE) [8,9].

Mathematical models of disease transmission are powerful tools for exploring this complex landscape of intervention strategies and quantifying the potential benefits of different options [10–13]. Traditional approaches in epidemiological modelling use compartmental models [14–16], which assume a uniform population and simple mixing patterns with steady contact rates. Such models can give qualitative answers for large-scale populations at best [17,18]; however, they are not suitable to account for the complexity and specifics of COVID-19 in local communities and small populations (e.g. hospital, workplace, school). Such settings are characterized by heterogeneous populations, multiple disease pathways and complex social interactions.

Our focus here is COVID-19 transmission in a hospital setting, where healthcare workers (HCWs) are at high risk to acquire infection through interactions with fellow HCWs and with patients [19–22]. We developed a novel individual-based model (IBM) for COVID-19 transmission among HCWs, and applied it to explore the efficacy of different control/mitigation strategies via non-pharmaceutical interventions. IBMs have been used extensively to model pathogens spread on different scales, from global pandemics [23–25] to local social networks [26]. On the disease side, our IBM features distinct infective stages and transitions, observed in COVID-19, with some hosts recovering without any symptoms, while others undergo mild or severe infection pathways. On the social side, we take into account individual behaviour, including mixing patterns among HCWs, their use of facemasks/PPE and HCW–patient interactions. All of these factors play an important role in COVID-19 transmission.

The IBM was calibrated in a Bayesian framework using empirical data from a non-COVID hospital unit. We used our calibrated model to simulate different intervention scenarios. In each case, we assessed the effect of interventions on outbreak outcomes: outbreak size and workday loss.

# 2. Material and methods

## 2.1. Individual-based modelling methodology

In our model, an individual can undergo a sequence of infection stages, classified as susceptible ($S$), pre-symptomatic/asymptomatic ($E$), two symptomatic stages $I_1$ (upper respiratory stage) followed by $I_2$ (advanced infection stage, lungs, etc.) and recovered/immune state ($R$) (figure 1). These states differ by their infectivity levels and stage duration. Unlike most other viral diseases, pre-symptomatic/asymptomatic COVID-19 hosts ($E$-stage) are known to transmit pathogens [27–29]. So, we assign positive infectivity levels ($b_0$, $b_1$, $b_2$) to all three stages ($E$, $I_1$, $I_2$).

We modelled social mixing patterns by assuming that HCWs and ward patients interact on a daily basis via aggregating in random groups of HCWs, and via patient visitation by HCWs (see electronic supplementary material, appendix for details). The net outcome is a contact pool for each HCW–host, which varies randomly on a daily basis. Each contact of a susceptible individual with infectious individuals (HCWs or patients) can lead to infection (transition $S \to E$), with a probability that depends on infectivity levels of the contact pool and the host susceptibility, $a$ ($a = 0$—fully protected, $a = 1$—fully susceptible). The latter depends on host health/immune status, individual behaviour, e.g. use of facial masks, and environmental conditions. For instance, HCWs are supposed to use additional protection when contacting patients. Then a probability of 'surviving' a single infective contact ($b_i$) for an $S$-host of susceptibility $a$, is given by $1 - ab_i$. Combining all infective contacts of a given $S$-host, we get the probability of infection ($S \to E$), $p_s = 1 - \prod_m (1 - ab_m)$, which depends on host susceptibility ($a$), contact pools ($m$) and contacts' infectivity ($b_m$), where $b_m$ could be 0 (susceptible or recovered), $b_1$ (asymptomatic) or $b_2$ (symptomatic). Note that contact rates, susceptibility and infectivity play different roles in the probability of infection, so we can explore them separately.

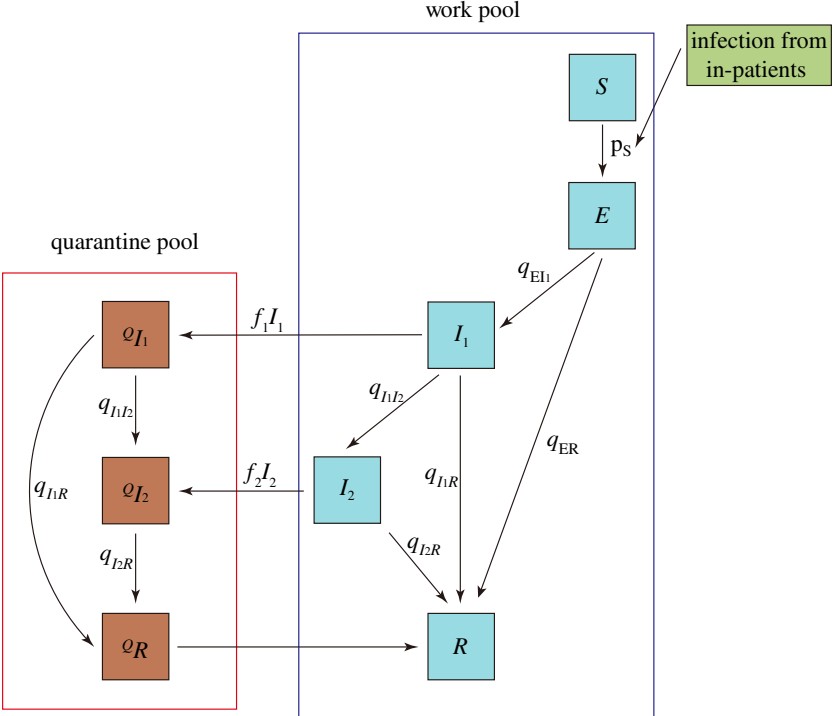

**Figure 1.** State transitions in the IBM. The standard SEIR scheme is used to describe host states: $S$, susceptible; $E$, latent (pre-symptomatic/asymptomatic but infectious); $I_1$, first symptomatic (upper respiratory infection); $I_2$, second stage (advanced lung infection); $R$, recovered/immune. Hosts can undergo three different pathways: asymptomatic ($E \rightarrow R$); mild symptomatic ($E \rightarrow I_1 \rightarrow R$); severe symptomatic ($E \rightarrow I_1 \rightarrow I_2 \rightarrow R$). Depending on the screening procedure, fractions of ($I_2$, $I_2$) are sent to quarantine, and released to the workpool upon recovery. Infected patients are treated as an external source.

We divided all HCW staff into susceptibility strata based on the hospital survey [30]: (i) normal pool, 60% of HCWs, have baseline susceptibility value, $a_N = 0.5$; and (ii) high-risk (stressed) pool, 40% of HCWs, with susceptibility level, $0.5 < a_S < 1$ (to be calibrated) (see electronic supplementary material, appendix for details).

Two points of our set-up require some clarification: (i) the proposed form of social mixing in random clusters extends the conventional 'social network' transmission pathways (see electronic supplementary material, appendix figures S1 and S2); (ii) an infective 'social contact' in our context means an event of sufficient duration and proximity, to allow transmission of pathogens from infected to susceptible host [31].

There is much uncertainty on disease progression of infective stages. Here, we assume infected $E$-hosts can undergo three different pathways: (A) asymptomatic ($E \rightarrow R$); (M) mild symptomatic ($E \rightarrow I_1 \rightarrow R$); (S) severe symptomatic ($E \rightarrow I_1 \rightarrow I_2 \rightarrow R$), with population fractions ($v_A$; $v_M$; $v_S$). In all cases, pre-symptomatic/asymptomatic pool ($E$) can carry and transmit the virus, along with ($I_1$, $I_2$). Each infective stage ($E$, $I_1$, $I_2$) has associated (mean) duration, $L_E$; $L_1$; $L_2$.

Specifically, the transition to the next stage is a random Bernoulli draw with success probability determined by a sigmoid function, $\Phi(d/L) = (d/L)^k / (1 + (d/L)^k)$, where $d$ is the time (days) that the host spent at a given disease stage and $L$ is the associated mean duration of the disease stage. When the steepness $k$ of the sigmoid function is high enough ($k > 6$), the mean duration of the Bernoulli probability is close to the associated mean duration, $L$ (see electronic supplementary material, appendix figures S5 and S6 for details).

During an outbreak, the HCWs expressing symptoms are tested, and certain fractions ($f_1$; $f_2$) of ($I_1$; $I_2$) are put in isolation, where they undergo their specific disease pathways, but do not mix and transmit the pathogen. Two different types of diagnostic tests were used in the hospital, PCR for light symptoms and lung-scan for more severe conditions [30,32]. Thus, our assumed quarantine fractions ($0 < f_i < 1$) account for limited test sensitivity, and a possible overlap of 'COVID-like' symptoms, expressed by non-COVID hosts. The recovered HCWs return to the work pool (figure 1).

The model simulations were run on a daily basis and implemented in the Wolfram Mathematica platform. The key inputs in the model include: (i) population make-up in terms of asymptomatic, mild and severe (A–M–S) progress groups; (ii) initial infection status of HCW pool; (iii) infectivity

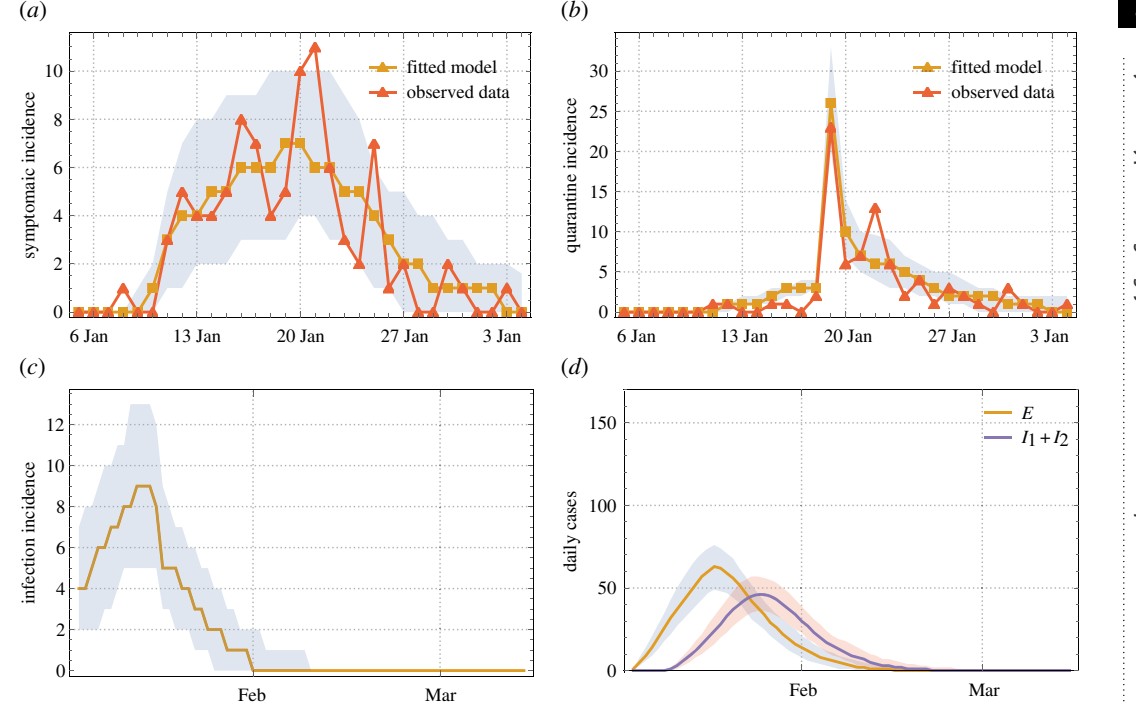

**Figure 2.** Model calibration and prediction. (*a*) The observed and fitted daily incidence of symptomatic cases ($E \rightarrow I_1$). (*b*) The observed and fitted daily quarantine cases among HCWs. (*c*) The predicted infection incidence ($S \rightarrow E$) from the calibrated model. (*d*) The corresponding predicted daily pre-symptomatic/asymptomatic cases, $E$, and symptomatic cases, $I_1 + I_2$, respectively. The grey shaded regions are 90% credible intervals.

levels ($b_0$, $b_1$, $b_2$) for ($E$, $I_1$, $I_2$) stages and susceptibility levels of individual hosts or host pools; (iv) average duration of infective stages for (A–M–S) pathway; (v) daily social mixing patterns between HCWs and infected patients; and (vi) daily isolation of symptomatic cases and recovery (see electronic supplementary material, appendix in details).

## 2.2. Calibration methodology

Our model is calibrated to empirical data of a COVID-19 outbreak among HCWs in the department of neurosurgery of Union Hospital, Wuhan, China, from 5 January to 4 February 2020 [30]. A Bayesian method is used to calibrate the following important parameters in our IBM: (i) mean infectivity ($b_1$, $b_2$) of symptomatic hosts ($I_1$, $I_2$), (ii) increased susceptibility level ($0.5 < a_S < 1$) of the high-risk pool; and (iii) fraction $v_A$ of HCWs going through the asymptomatic pathway ($E \rightarrow R$).

The Bayesian method uses the posterior probability distribution to quantify the uncertainties in these model parameters using the observed data on the daily incidence of symptomatic cases and the daily isolated cases (figure 2). The prior distributions for all these parameters are taken to be uniform within acceptable ranges (see electronic supplementary material, appendix figure S3). The likelihood for the observed data is assumed as a normal distribution with the centre at the predicted values from the IBM. The adaptive Metropolis algorithm [33] is used to sample from the posterior distribution, where the jump size is adaptively chosen based on the sample covariances. The chains are run for 10 000 iterations, and after 5000 burn-in every 50th sample is used as the final sample from the posterior distribution. To assess the convergence of the posterior sampling, the Gelman–Rubin statistic [34] is computed for all the parameters. The statistics are found to be very close to 1, the desired value in strong support of convergence. The calibration was implemented using R statistical software.

## 2.3. Dataset from Wuhan hospital outbreak

On 26 December 2019, a patient later diagnosed with COVID-19 was admitted in the department of neurosurgery of Union Hospital, Wuhan, China. No PPE was used by HCWs at that time. By 8 January, HCWs started to show COVID-like symptoms (headache, cough, sore throat), and screening

and isolation were initiated among HCWs. From 19 January, patient admission was stopped in the department, and the hospitalized patient pool was gradually reduced from 200 to 20 by the beginning of February. Over the period from 5 January to 4 February, 92 out of the 171 HCWs of the department were suspected or confirmed COVID-19 cases and isolated. New patients were only admitted in early March 2020 when the pandemic was declared under control in Wuhan.

## 2.4. Intervention strategies

We consider three types of interventions: (i) social distancing (reduced contact rates) and individual protection (facial masks) among HCWs, (ii) enhanced screening and isolation of infected HCWs, and (iii) patient-pool control (pool size and infection level), and individual HCW protection via PPE for HCW–patient interaction.

To assess the strategies for mitigating the COVID-19 outbreak using the calibrated model, we want to consider a baseline case that is realistic, simple and general (50 and 100% isolation (quarantine) fractions of symptomatic cases ($I_1$; $I_2$), respectively, and fixed infection level of the patient pool (i.e. 200 patients pool with 2% infected). To account for model uncertainties, we run each control simulation for 100 posterior parameter samples and five stochastic model realizations for each sample (500 histories altogether), over a six-month period.

For social distancing, we considered 50 and 75% contact-rate reduction relative to their baseline values. The effect of facemask on inter-staff or staff–patient mixing was simulated by reduced susceptibility of individual HCWs, with several values of mask efficacy [9]. Screening and isolation fractions ($f_i$) of HCWs were based on limited test sensitivity, combined with non-COVID symptoms. An increase in targeted isolation assumes more intensive screening or test sensitivity. We also studied the effect of isolating pre-symptomatic/asymptomatic cases ($E$ pool). This task is more challenging, as PCR tests have lower sensitivity for such hosts [32], so to identify a suitable $E$ fraction would require intensive mass screening or contact tracing.

For quantitative assessment of control interventions and their impact, we use two measures: (i) outbreak size = infection turnover (by the end of outbreak); (ii) workday loss estimated from the quarantine pool over the outbreak duration. The latter gives a simple economic measure of outbreak impact and putative interventions. In each control experiment, we compare the ratio of two outputs (outbreak size and workday loss) to their baseline values, and record these relative values and their distribution.

Another important factor in the hospital setting is the in-patient pool. In our case (a non-COVID unit in Wuhan), it varied from the full capacity to zero. The key inputs of the patient pool included (i) infection prevalence and (ii) mean patient infectivity to HCWs. The former is controlled by patient admission and screening/isolation procedures; the latter can be modulated by using PPE. We also explored the effect of different timing of PPE implementation and its efficacy.

# 3. Results

## 3.1. Model calibration with hospital data

The predictions from the calibrated IBM were very close to the observed data on daily symptomatic and quarantine cases (figure 2). The fraction of asymptomatic disease-progress pool, $v_A$, was estimated at 0.31 (90% credible interval (CrI): 0.16–0.40). So, a sizable part of transmission was carried over by undetected cases ($E$ pool). Susceptibility level of the high-risk pool was estimated at $a_S = 0.76$ (90% CrI: 0.58–0.97). We attributed a higher susceptibility level to work stress, and our results gave a quantitative measure to this increase at 52% (90% CrI: 16.4–93.0%) above the normal level. The infectivity levels of pre-symptomatic and symptomatic infections were estimated to be 0.12 (90% CrI: 0.11–0.14) and 0.23 (90% CrI: 0.20–0.26). See electronic supplementary material, appendix figure S3 for the prior and posterior probability distributions.

## 3.2. Analysis of interventions

The baseline scenario showed that almost all HCWs get infected, resulting in significant workday loss, 1050 (90% CrI: 913–1282), over the six-month period (table 1; electronic supplementary material, appendix figure S4). The impact of implementing social distancing through reduction of contact rates

**Table 1.** Effects of implementing social distancing through reduction of contacts alone and wearing face masks alone, from the start of the outbreak. We simulated a six-month intervention-regimen for the calibrated model. The progress was measured in terms of outbreak size, workday loss and cumulative quarantine incidence. Reasonable (50 and 75%) reduction of contact rates and levels of efficacy of facial masks (50, 67, 75, 85 and 95%) were chosen. The results shown are predicted median (90% credible interval).

| intervention | | | | | | | | |
| --- | --- | --- | --- | --- | --- | --- | --- | --- |
| | | reduction of contact rates | | efficacy of facial masks | | | | |
| progress measure | baseline results (no intervention) | 50% | 75% | 50% | 67% | 75% | 85% | 95% |
| outbreak size | 170 (168–171) | 164 (159–169) | 146 (135–156) | 159 (150–165) | 141 (128–151) | 124 (108–137) | 90 (75–104) | 35 (24–46) |
| workday loss | 1050 (913–1282) | 992 (855–1202) | 853 (721–1044) | 950 (817–1145) | 819 (678–997) | 698 (562–876) | 469 (342–605) | 142 (77–212) |
| cumulative quarantine incidence | 112 (98–136) | 107 (93–130) | 93 (80–113) | 103 (89–123) | 90 (76–108) | 78 (63–96) | 53 (40–68) | 16 (9–24) |

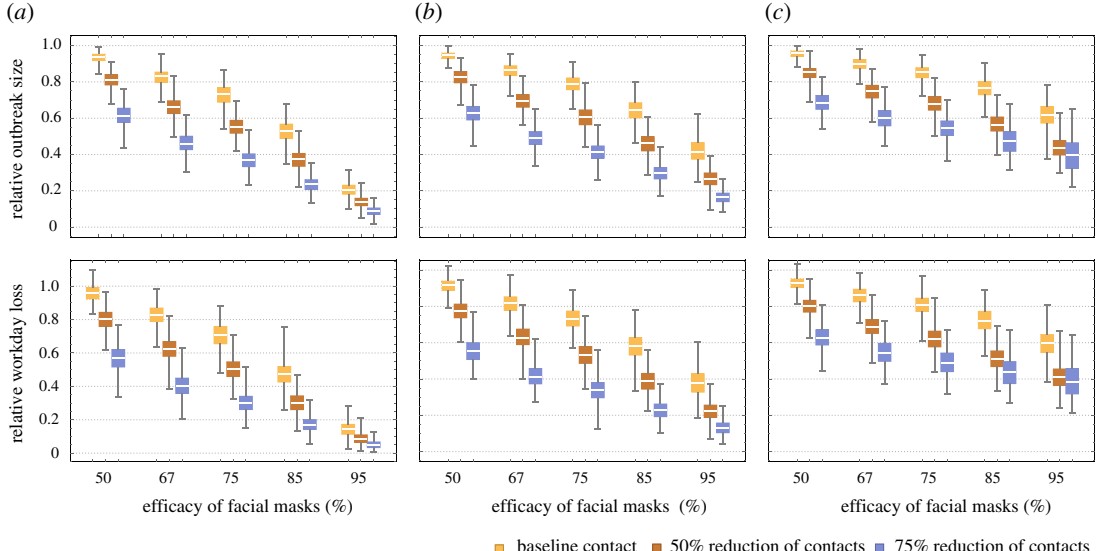

**Figure 3.** The combined effect of facial masks and social distancing. Three levels of social distancing were considered (normal, reduced by 50%, reduced by 75%). For facemask efficacy, we considered five putative values (50, 67, 75, 85 and 95%). The efficacy is measured via reduced host susceptibility per contact ($a \rightarrow 0.5 * a$; $a \rightarrow 0.33 * a \ldots$). We also considered different timing of preventive measures: (i) start of the outbreak ($a$); (ii) after the first identified HCW case ($b$); and (iii) after 10% of HCW staff got infected ($c$). In each case, we estimated the posterior distribution of the relative outbreak size, and the workday loss over baseline values.

alone and wearing facemasks alone, from the start of the outbreak, was evaluated (table 1). The reduction of contact rates alone has a marginal effect on mitigating the outbreak in the long run. The 50% drop of contact rates leads to about 4–6% reduction of the outbreak size and workday loss, while 75% drop leads to a 15–17% reduction, relative to baseline values. The efficacy of facemasks is uncertain, and we explored several values (50, 67, 75, 85 and 95%), based on the previous studies [9]. We have shown that wearing facemasks had a higher impact on mitigating the outbreak, than social distancing (reduced contact rates). At 95% efficacy, we could achieve 80% (90% CrI: 73.1–85.7%) reduction of outbreak size, and 87% (CrI: 80.0–92.5%) of workday loss, compared to the baseline.

Figure 3 illustrates the combined effect of facemask and social contact. We used the same values of facemask efficacy and contact rates as table 1. For each value of facemask efficacy, we observed a consistent reduction of the outbreak size with reduced contact rates. It varied from 13 to 34% drop for low-efficacy facemask (50% protection) to 30 to 60% drop for high-efficacy facemask (95% protection). We observed a similar percentage reduction for the workday loss. So, the impact of reduction of contact rates was much greater under the higher efficacy of facemasks.

We also explored the effect of timing of intervention by the following three scenarios: (i) at the beginning (figure 3$a$); (ii) after the first identified case (figure 3$b$); and (iii) after 10% of HCWs have been identified as infected (figure 3$c$). Early interventions have made marked improvement under different types of facemasks and contact rates. For instance, if control interventions (adoption of high-efficacy facemasks and reduced contact rates) were implemented at the beginning, we observed 80–90% reduction of the outbreak size (a near-complete control). A later implementation (e.g. after the first identified case) gave 60–85% reduction. If the timing was delayed to, for example, 10% identified cases, these numbers dropped to 40–60%. All intermediate cases are shown in figure 3.

We next looked at the effect of HCWs screening and isolation via two scenarios. The first scenario considered symptomatic cases only, by changing quarantine fraction ($f_1$) of $I_1$, from its baseline value (50%) to 60−100%; quarantine fraction ($f_2$) of $I_2$ was fixed at 100%. Figure 4$a$ shows increased symptomatic isolation had only a marginal effect on the outbreak size, while raising workday loss. A clue to low efficacy of symptomatic screening lies in (i) the role of pre-symptomatic/asymptomatic ($E$) pool in transmission and (ii) contribution of the patient source. To test (i), we extended our quarantine strategy to $E$ pool. Of course, such an extension requires intensive screening of the work pool. Under random selection, isolating $f$ fraction of $E$ would require much more than $f$ fraction of HCWs tested. For numeric simulations, we fixed ($f_1$, $f_2$) at (90, 100%), and varied $E$ fraction from 10 to 60%. We still found the effect of such a strategy was limited, it often prolongs the outbreak duration

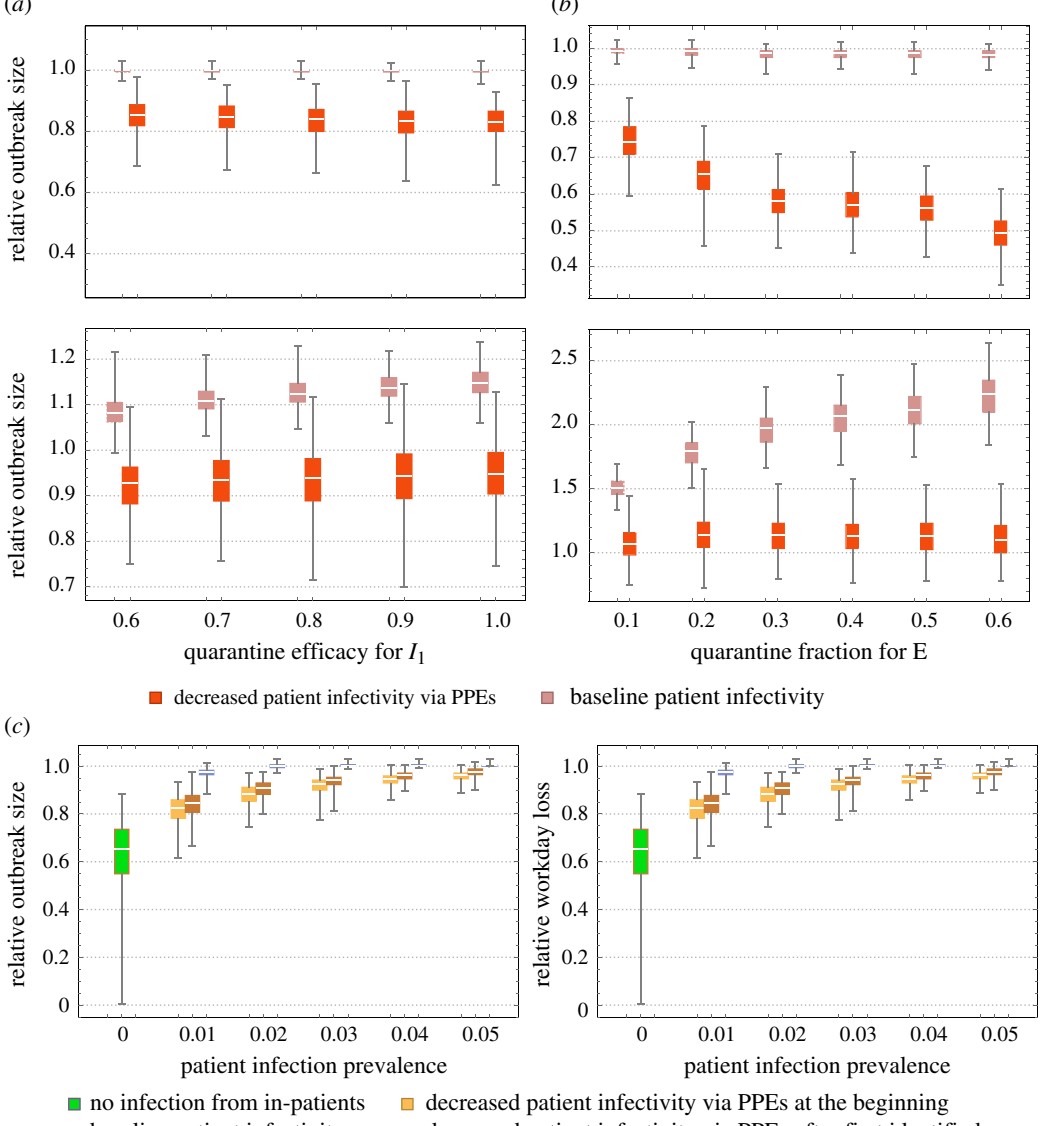

**Figure 4.** The effect of the quarantine and patient sources on relative outbreak size and workday loss. The patient sources (infection prevalence and infectivity) were controlled via screening/isolation and the use of PPE by HCWs. We considered two quarantine strategies for HCWs: symptomatic cases only (*a*) and adding pre-symptomatic/asymptomatic cases (*b*). We used the following marking: (pink) baseline patient infection level, (red) reduced patient infection by 80% via PPE use by HCWs. In (*a*), the quarantine fraction of moderate/severe cases ($I_2$) was fixed at 100%, and the quarantine fraction of mild cases ($I_1$) was varied from 60 to 100%. In (*b*), we fixed symptomatic ($I_1$; $I_2$) quarantine fractions at (90%, 100%), and varied the quarantine fraction of pre-symptomatic/asymptomatic *E* pool from 10 to 60%. (*c*) The effect of patient infection and different timing of PPE use: (i) start of the outbreak, (ii) after the first identified HCW infection and (iii) no PPE use. We considered different levels of prevalence of the infected patient pool: 0, 1, 2% (baseline value), 3, 4, 5%.

without affecting its size. Besides, such a strategy can incur an economic burden by increased workday loss, though the effect is subtler, as increased quarantine rate can slow the transmission rate; hence, fewer hosts would be infected and need isolation. More significant progress was achieved by controlling the patient source, via reduced patient prevalence (screening), or reduced infectivity (PPE) (figure 4*a*,*b*).

We ran several experiments with patient-pool control and PPE use (figure 4*c*). For PPE timing, we made three choices: (i) the start of an outbreak, (ii) after the first identified HCW case, and (iii) no PPE use. We assumed PPE provides 80% protection (via the reduced probability of transmission from an infected patient). We also varied the infected prevalence level of the patient pool, from 0 to 5% (baseline case was 2%). We found the control of patient infection (via e.g. PPE, screening and isolation, particularly for new patients) can reduce outbreak size, even though the bulk of transmission is carried over by inter-staff HCW contacts. We found the combined strategy (enhanced HCW screening/isolation with patient

control) could lead to marked improvement both in outbreak size and in workday loss. This effect, however, is not observed for quarantine alone, under persistent patient source.

Overall, we saw high-efficacy facemasks could provide the most effective control tool for reducing COVID-19 transmission in HCW staff (figure 3).

# 4. Discussion

With the spread of COVID-19 in the world, the development of strategies for mitigating its severity is a top public health priority. Large-scale population-level models of SARS-CoV-2 transmission can give some qualitative answers for outbreak control on regional/country scales [35]; however, few studies have looked at the effects of interventions in a local community setting, such as hospital, workplace and school.

Using a novel individual-based modelling approach, we explored different scenarios for COVID-19 transmission and control in a non-COVID hospital unit. Our IBM methodology employed conventional SEIR disease stages with graded infectivity, extended to heterogeneous host make-up, which includes multiple disease pathways, varying individual susceptibility and behavioural patterns. These factors can be affected by work stress, health status, use of face masks/PPE and social interactions. Detailed data on the COVID-19 outbreak in the department of neurosurgery of Union Hospital in Wuhan (China) [30] were used to calibrate the essential model parameters.

One of the key uncertain parameters was the pre-symptomatic/asymptomatic fraction, which was estimated at 31%, indicating a relatively high proportion of undetectable infections. We also estimated the infectivity levels of pre-symptomatic ($E$) and symptomatic disease ($I$) states to be 0.12 and 0.23, respectively. Another uncertain input was individual susceptibility, which could be affected by health status or work stress. We estimated the high-risk susceptibility level relative to normal susceptibility and found work-related stress could increase the risk of COVID-19 infection by up to 52%.

The calibrated model was used to simulate a range of intervention scenarios, aimed at mitigating the outbreak and examining its impact on the work pool. The baseline case, without interventions, gave a large outbreak size, whereby almost all HCWs were infected over two months. It also incurred a significant workday loss for the unit. Such results support early modelling findings of large-scale populations, and subsequent empirical observations, that in the absence of control measures, a COVID-19 epidemic could quickly overwhelm a region [12]. High-efficacy facemasks were shown to be most effective for reducing infection cases and workday loss. The impact of social distancing through the reduction of contact rates alone had a marginal effect on mitigating the outbreak in the long run. Reducing social contact rates to 50% (or 75%) resulted in a 4–6% (or 15–17%) drop in the outbreak size, and a similar drop in the workday loss, compared to the baseline case. However, the impact of reduction of contact rates was much greater under the higher efficacy of facemasks.

Implementing the quarantine policy (HCW screening and isolation) alone, even when all symptomatic cases are included, would typically prolong the outbreak duration, but had a marginal effect on its size, particularly under the external (patient) source pressure. Our results indicated that the low efficiency of symptomatic quarantine was due to a large share of transmission being carried by pre-symptomatic/asymptomatic ($E$) individuals [36], and to the patient source. Our results also showed that a quarantine policy for HCWs should be augmented with other interventions to achieve a significant reduction. Efficient control of the patient source (via the use of PPE, their screening and isolation, and/or admission) is one key to mitigating the HCW outbreak. The effectiveness of all these interventions was shown to increase with their early implementation.

To our knowledge, this study is the first of its kind to provide quantitative modelled assessment and projections for COVID-19 transmission in hospital settings. However, the IBM methodology developed here has a far broader scope, beyond healthcare facilities. Indeed, with proper adjustment, it could be applied to many other local communities (workplaces, schools, city neighbourhoods, etc.). The key feature of such IBM is a fine-scale resolution of community make-up, social interactions and disease pathways. Such information is essential for risk assessment and the development of efficient control/intervention strategies on a local scale.

The current model set-up is subject to some limitations. First, it was designed for a single hospital unit and simplified treatment of the patient pool, as the target group in our study was HCW pool. More realistic local communities could combine multiple units (e.g. large hospital), with refined population structure (e.g. patients, visitors, staff), and more complex interactions (e.g. 'random' and 'scheduled' contact pools). Empirical data on these interactions will be required to adequately parametrize such

models. Second, although we have made an effort to characterize the SARS-CoV-2 transmission in a hospital setting, some parameters used in our set-up were drawn from general information sources, such as fractions of symptomatic mild and severe cases [37], disease stages and durations [38], and associated infectivity levels [39], which may be adjusted in the future work.

## 5. Conclusion

Overall, our analysis shows that a COVID-19 outbreak among HCWs in a non-COVID-19 hospital unit can be efficiently controlled/mitigated by non-pharmaceutical means. The most crucial factor of success is high-efficacy facemasks for HCW contacts. It can be further augmented by social distancing, screening/isolation and patient source control.

Ethics. The study protocol was approved by the institutional ethics board of Union Hospital, Tongji Medical College, Huazhong University of Science and Technology, Wuhan, China (no. 20200029). Written informed consent was required before the data collecting, and participants were informed that they could refuse to answer any question. The questionnaire did not ask about infection status, and no biological samples were collected.

Data accessibility. Data and relevant code for this research work are stored in GitHub: https://github.com/qimin-h/COVID-19-huang-et-al.- and have been archived within the Zenodo repository (https://doi.org/10.5281/zenodo.4122370).

Authors' contributions. Q.H., A.M. and X.J. contributed equally and shared the first authorship. Q.H., D.G., A.M. and M.N.-M. designed research, did model development, calibrations and simulations, X.J. H.Z., F.F., P.F. and X.W. collected and provided hospital data, Q.H. wrote the first draft, M.N.-M., D.G., A.M. and M.A.H. made critical revision of the manuscript. All the authors contributed to the interpretation of the study results, read, commented and approved the final version.

Competing interests. We declare we have no competing interests.

Funding. This work was supported by the National Science Foundation RAPID Award (grant no. DEB-2028631 to Q.H., A.M. and D.G.), the National Science Foundation RAPID Award (grant no. DEB-2028632 to M.N.-M.] and the Fundamental Research Funds for the Central Universities (grant no. 2020kfyXGYJ010 to X.J.); funders had no role in study design, data collection, data analysis, writing of the report or the decision to submit for publication. The corresponding authors had full access to all of the data and the final responsibility to submit for publication.

Acknowledgements. The authors would like to thank all healthcare workers in this study. X.J. and H.Z. had full access to all the data in the study and took responsibility for the integrity of the data. The authors would also like to thank the handling editor and reviewers for their helpful comments and suggestions, which improved the presentation of the manuscript.

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
