## [Peer Review File · Royal Society Open Science]

Review History

RSOS-201895.R0 (Original submission)

Review form: Reviewer 1

Is the manuscript scientifically sound in its present form?

Yes

Are the interpretations and conclusions justified by the results?

Yes

Is the language acceptable?

Yes

Do you have any ethical concerns with this paper?

No

Have you any concerns about statistical analyses in this paper?

No

Recommendation?

Major revision is needed (please make suggestions in comments)

Comments to the Author(s)

This is a nice modeling study of hospital transmission of COVID-19 among HCWs using an individual based model calibrated to data from Wuhan, China outbreak. The authors quantify sensitivity of outbreak size to a few intervention strategies and their cost, showing the importance of face-mask usage and limitations of screening and isolation due to asymptomatic transmission. However there are a few questionable model assumptions. Also the structure of the model, data, and parameter values may have identifiability issues or need clarification in some parts. Nevertheless, with some revisions, it can provide a worthwhile contribution for publication. Comments are below:

- 1) The usage of hospital data to inform on HCW susceptibility and contact patterns was not adequately explained. What is the structure of the data used? Does the data just give contact rate or also the distribution of contact "event" size (e.g. pairwise)?
- 2) The probability of transition between infected classes is somewhat non-standard, using a sigmoid function of the ratio between time spent and mean duration. Does this correspond to any known probability distribution for residence time that might be supported by data? Does this allow for duration larger than mean value? This should be clarified.
- 3) In methods or supplementary, the authors do not fully explain the data which the model was calibrated with. Are quarantine necessarily infected (or are some just potentially infected through exposure)? Do all symptomatic reported cases become quarantined? Related to this, is how the screening probabilities, f , are chosen from hospital policy data? Also unclear was how patient numbers, infection rate, contacts with HCW, were estimated from data?
- 4) What is the essential difference in the model between contact reduction and facemask usage since they seem to be both factor into transmission probability? Explaining this can help shed light on your finding that facemask use was more efficacious than contact reduction.
- 5) In Intervention section, it is stated "For our baseline case, we assumed 50% and 100% isolation (quarantine) fractions...". In the table, it seems f_1 and f_2 were chosen differently. Can you clarify what is meant for the baseline case?
- 6) Does the IBM have time step of 1 day? Why not consider a simpler Markov chain model?
- 7) The large number of parameters and closeness of datasets (it is not clear if reported cases is much different than quarantined since screening probabilities are not known) may cause identifiability problems. Perhaps you can more clearly point out how you address this issue.

Review form: Reviewer 2

Is the manuscript scientifically sound in its present form?

Yes

Are the interpretations and conclusions justified by the results?

Yes

Is the language acceptable?

Yes

Do you have any ethical concerns with this paper?

No

Have you any concerns about statistical analyses in this paper?

No

Recommendation?

Accept with minor revision (please list in comments)

Comments to the Author(s)

See attached file (Appendix A).

Decision letter (RSOS-201895.R0)

Dear Dr Huang

The Editors assigned to your paper RSOS-201895 "SARS-CoV-2 transmission and control in a hospital setting: an individual-based modelling study" have now received comments from reviewers and would like you to revise the paper in accordance with the reviewer comments and any comments from the Editors. Please note this decision does not guarantee eventual acceptance.

Please submit your revised manuscript and required files (see below) no later than 21 days from today's (ie 16-Feb-2021) date. Note: the ScholarOne system will 'lock' if submission of the revision is attempted 21 or more days after the deadline. If you do not think you will be able to meet this deadline please contact the editorial office immediately.

on behalf of Dr Shigui Ruan (Associate Editor) and Glenn Webb (Subject Editor)
openscience@royalsociety.org

Associate Editor Comments to Author (Dr Shigui Ruan):

Please revise your manuscript by following all comments and suggestions and provide a detailed itemized response.

Reviewer comments to Author:

Reviewer: 1

Comments to the Author(s)

This is a nice modeling study of hospital transmission of COVID-19 among HCWs using an individual based model calibrated to data from Wuhan, China outbreak. The authors quantify sensitivity of outbreak size to a few intervention strategies and their cost, showing the importance of face-mask usage and limitations of screening and isolation due to asymptomatic transmission. However there are a few questionable model assumptions. Also the structure of the model, data, and parameter values may have identifiability issues or need clarification in some parts. Nevertheless, with some revisions, it can provide a worthwhile contribution for publication. Comments are below:

- 1) The usage of hospital data to inform on HCW susceptibility and contact patterns was not adequately explained. What is the structure of the data used? Does the data just give contact rate or also the distribution of contact "event" size (e.g. pairwise)?
- 2) The probability of transition between infected classes is somewhat non-standard, using a sigmoid function of the ratio between time spent and mean duration. Does this correspond to any known probability distribution for residence time that might be supported by data? Does this allow for duration larger than mean value? This should be clarified.
- 3) In methods or supplementary, the authors do not fully explain the data which the model was calibrated with. Are quarantine necessarily infected (or are some just potentially infected through exposure)? Do all symptomatic reported cases become quarantined? Related to this, is how the screening probabilities, f , are chosen from hospital policy data? Also unclear was how patient numbers, infection rate, contacts with HCW, were estimated from data?
- 4) What is the essential difference in the model between contact reduction and facemask usage since they seem to be both factor into transmission probability? Explaining this can help shed light on your finding that facemask use was more efficacious than contact reduction.
- 5) In Intervention section, it is stated "For our baseline case, we assumed 50% and 100% isolation (quarantine) fractions...". In the table, it seems f_1 and f_2 were chosen differently. Can you clarify what is meant for the baseline case?
- 6) Does the IBM have time step of 1 day? Why not consider a simpler Markov chain model?
- 7) The large number of parameters and closeness of datasets (it is not clear if reported cases is much different than quarantined since screening probabilities are not known) may cause identifiability problems. Perhaps you can more clearly point out how you address this issue.

Reviewer: 2

Comments to the Author(s)

See attached file.

===PREPARING YOUR MANUSCRIPT===

===PREPARING YOUR REVISION IN SCHOLARONE===

Author's Response to Decision Letter for (RSOS-201895.R0)

See Appendix B.

Decision letter (RSOS-201895.R1)

Dear Dr Huang,

It is a pleasure to accept your manuscript entitled "SARS-CoV-2 transmission and control in a hospital setting: an individual-based modelling study" in its current form for publication in Royal Society Open Science.

COVID-19 rapid publication process:

We are taking steps to expedite the publication of research relevant to the pandemic. If you wish, you can opt to have your paper published as soon as it is ready, rather than waiting for it to be published the scheduled Wednesday.

This means your paper will not be included in the weekly media round-up which the Society sends to journalists ahead of publication. However, it will still appear in the COVID-19 Publishing Collection which journalists will be directed to each week (<https://royalsocietypublishing.org/topic/special-collections/novel-coronavirus-outbreak>).

If you wish to have your paper considered for immediate publication, or to discuss further, please notify openscience_proofs@royalsociety.org and press@royalsociety.org when you respond to this email.

on behalf of Dr Shigui Ruan (Associate Editor) and Glenn Webb (Subject Editor)
openscience@royalsociety.org

Appendix A

Review of

SARS-CoV-2 transmission and control in a hospital setting: an individual-based modelling study

Q. Huang, A. Modal, X. Jian, M. Horn, F. Fan, P. Fu, X. Wang, H. Zhao, M. Ndeffo-Mbah, and D. Gurarie

Submitted to *Royal Society Open Science*

This paper examines the extent of COVID-19 infections in a non-COVID-19 hospital setting. The goal is to understand how COVID-19 outbreaks among health-care workers (HCW) in these settings can be reduced by non-pharmaceutical measures. The hospital setting involves patient and HCW populations (other populations, such as visitors and staff are not included). These populations are subdivided into S (individuals susceptible to infection), E (pre-symptomatic or asymptomatic infectious individuals), I1 (first stage symptomatic infectious individuals), I2 (advanced stage symptomatic infectious individuals), and R (removed or immune infected individuals). The epidemic outbreak pathway in this setting depends on (1) daily contacts between and among patients and HCW, (2) levels of infectivity moderated by facemasks, proximity, timing of visits, and other protective measures, and (3) testing and isolation based quarantine measures.

The analysis uses an individual-based modeling (IBM) methodology. A Bayesian method was used for the identification of model parameters. The model parameters were based on calibrated data from a COVID-19 outbreak in the department of neurosurgery of Union Hospital in Wuhan, China from January 5th, 2020 to February 4th, 2020. Three intervention strategies were considered: (i) social distancing (reduced contact rates) among HCW and individual protection (facial masks); (ii) enhanced screening and isolation of infected HCW; and (iii) patient-pool control (pool size and infection level control), and individual protection via HCW personal protection equipment. The Wuhan hospital data and the model output produced a highly significant correlation. The analysis of the intervention scenarios revealed that reduction of contact rates alone had marginal impact, and the wearing of facemasks had much higher impact for reducing the number of infectious cases and the number of workday losses. Also, HCW quarantine alone only prolonged the outbreak duration.

This study is a most valuable contribution to understanding the impact of COVID-19 on HCW, which is of great importance in the toll of COVID-19 on society. The method of study, using an IBM calibrated to an actual hospital data set, is novel, and possibly adaptable to other settings (such as schools, neighbors, and workplaces). This is an excellent work and should be accepted for publication with very minor corrections.

Corrections:

On line 136, January 5th, 2019 should be changed to January 5th, 2020.

There is some confusion in the descriptions on lines 162 -163, between the distinction of (i) and (iii).

Update Reference (30) to “Risk factors of SARS-CoV-2 infection in healthcare workers: a retrospective study of a nosocomial outbreak”, Wang X, Jiang X, Huang Q, Wang H, *et al.*, *Sleep Medicine X 2* (2020) 100028.

Update other references

Appendix B

Responses to the comments and suggestions by the reviewers on Manuscript ID RSOS-201895

Reply to reviewer #1:

This is a nice modeling study of hospital transmission of COVID-19 among HCWs using an individual based model calibrated to data from Wuhan, China outbreak. The authors quantify sensitivity of outbreak size to a few intervention strategies and their cost, showing the importance of face-mask usage and limitations of screening and isolation due to asymptomatic transmission. However, there are a few questionable model assumptions. Also the structure of the model, data, and parameter values may have identifiability issues or need clarification in some parts. Nevertheless, with some revisions, it can provide a worthwhile contribution for publication. Comments are below:

1. The usage of hospital data to inform on HCW susceptibility and contact patterns was not adequately explained. What is the structure of the data used? Does the data just give contact rate or also the distribution of contact "event" size (e.g. pairwise)?

Response: We thank the reviewer for pointing this out. We have added a new section in Appendix to explain the usage of hospital data in detail.

Susceptibility depends on many factors such as health /immune status, individual behavior e.g., use of facial masks, and environmental conditions. In our previous study [30], we found that poor sleep quality and working under pressure may increase the risk of nosocomial SARS-CoV-2 infection among HCWs. As described in our paper [30], an online electronic questionnaire was sent to all 171 HWCs in the Department of Neurosurgery of Union Hospital of Wuhan, and 118 valid questionnaires were finally collected. Among 118 HCWs, about 40% indicated that they worked under pressure. So we simply divided all HCWs staff into susceptibility strata: (i) normal pool, 60% of HCWs, have an arbitrary baseline susceptibility values $a_N = 0.5$; (ii) high-risk (stressed) pool, 40% of HCWs, with susceptibility level, $0.5 < a_s < 1$ (to be calibrated). We have now clarified it clearly in the Appendix.

The data we used in this manuscript are the daily number of symptomatic cases and quarantine cases from Jan 8th, 2020 to Feb 3rd, 2020, and other general information from hospitals, such as approximate mean contact rate between HCWs, the total number of patients, the mean number of visitations to patient per HCW, and the quarantine fraction of HCWs. The data doesn't give the distribution of contact "event" size, we generate such realistic contact "event" size (60 pair-contacts, 30 triple-contacts, 8 quadruple-contacts per day) to approximately give a 2.2 contact rate per HCW-host per day, which is consistent with the mean contact rate that hospital authority provides us. We have now clarified it clearly in the Appendix.

2. The probability of transition between infected classes is somewhat non-standard, using a sigmoid function of the ratio between time spent and mean duration. Does this correspond to any known probability distribution for residence time that might be supported by data? Does this allow for duration larger than mean value? This should be clarified.

Response: Indeed, a common method is to use the inverse of mean duration as the probability of transition. Since our individual-based model is able to track days an individual spent in each infection stage, our approach via the sigmoid function could provide a more reasonable way to model the probability of transitioning to the next stage. Here the probability increases as time spent in the given stage increases. Note that the transition to the next stage follows a Bernoulli probability distribution with success probability q , where q is the sigmoid function that depends on time spent and the mean duration. Such transition allows the duration to be larger than the mean value. We have now added a detailed explanation and Figures S5-S6 to illustrate our approach and clarified it accordingly in the main manuscript with more details in the Appendix.

3. In methods or supplementary, the authors do not fully explain the data which the model was calibrated with. Is quarantine necessarily infected (or are some just potentially infected through exposure)? Do all symptomatic reported cases become quarantined? Related to this, is how the screening probabilities, f , are chosen from hospital policy data? Also unclear was how patient numbers, infection rate, contacts with HCW, were estimated from data?

Response: We thank the reviewer for pointing these out. We used the observed data on the daily newly symptomatic cases and the daily newly isolated cases to do model calibration which is mentioned in the main article (Figure 2).

The quarantine was done if a healthcare worker tested positive or had covid-like symptoms. Please note that the outbreak that occurred in this hospital unit was at the very early stage of the pandemic, when people had little awareness of the transmission of COVID-19. Initially the fraction of symptomatic HCWs that were quarantined was very low but after January 19 that fraction increased dramatically due to hospital policy change. Still the fraction was not 100% as there was some delay in quarantining a few cases. The assumed quarantine fractions ($0 < f_i < 1$) combine limited test sensitivity, and the delay in testing and quarantining. The screening probabilities before and after January 19 was provided by the hospital authority what they believe are reasonable according to their policy and records. As the reviewer correctly pointed out (Comment No 7.) that too many unknown parameters may cause identifiability issues, so we fixed the parameters that can be reasonably provided by the hospital.

The patient numbers were provided by the hospital records, the average contacts with HCWs are also provided by the hospital which reflects the day-to-day activities. The patient infection rate (0.02) was consistent with the hospital record. We have now added an additional paragraph on the full data description in the Appendix.

4. What is the essential difference in the model between contact reduction and facemask usage since they seem to be both factor into transmission probability? Explaining this can help shed light on your finding that facemask use was more efficacious than contact reduction.

Response: The probability of infection $p_s = 1 - \prod_m (1 - ab_m)$ of a susceptible individual in our methodology depends on his/her susceptibility (a), contact pool m , and his/her contacts' infectivities (b_m), where b_m could be 0 (susceptible or recovered), or b_1 (asymptomatic) or b_2 (symptomatic). Note that contact rates, susceptibility, and infectivity play different roles in the probability of infection, so we can explore them separately. Wearing a facemask results in reducing the parameter a and implementing contact reduction results in reduced contact pool m , respectively. We have now clarified it in the manuscript.

Table 1 illustrates the effects of implementing social distancing through reduction of contacts alone and wearing face masks alone, from the start of the outbreak. It shows that wearing high-efficacy facemasks was more efficacious than contact reduction.

5. In Intervention section, it is stated "For our baseline case, we assumed 50% and 100% isolation (quarantine) fractions...". In the table, it seems f_1 and f_2 were chosen differently. Can you clarify what is meant for the baseline case?

Response: In Table S1, The screening probabilities f_1 and f_2 before and after January 19 was provided by the hospital authority what they believe are reasonable according to their policy and records, where initially the fraction of symptomatic HCWs that were quarantined was very low but after January 19 that fraction increased dramatically due to hospital policy change. We used these known parameter values as given in Table S1 for our model calibration.

When we assess the strategies for mitigating the COVID-19 outbreak using the calibrated model, we want to consider a baseline case that is realistic, simple, and general (50% and 100% quarantine fractions for I_1 and I_2 , respectively). We have now clarified it both in the Manuscript and in the Appendix.

6. Does the IBM have time step of 1 day? Why not consider a simpler Markov chain model?

Response: Exactly, the IBM has a time step of 1 day, which is a natural unit for human behavior and disease transmission.

IBM allows greater flexibility and individual details of host behavior and disease progress (i.e., a heterogeneous host population with distinct infective stages and transitions, different patterns of disease-progression (asymptomatic, mild, severe), individual risk factors (health status, work stress, et al), individual behavior including social mixing patterns among HCWs, use of PPE/ facial masks, and HCW-patient interactions) in a hospital setting. All those are important for Covid-19 transmission. A simpler Markov chain model may not be able to take into account these important factors.

7. The large number of parameters and closeness of datasets (it is not clear if reported cases are much different than quarantined since screening probabilities are not known) may cause identifiability problems. Perhaps you can more clearly point out how you address this issue.

Response: Indeed, too many unknown parameters may cause identifiability issues, so we fixed the parameters that could be reasonably provided by the hospital or obtained from published literature. We calibrated a minimal number (3) of key parameters: (i) mean infectivity of symptomatic hosts, (ii) increased susceptibility level of the high-risk pool; (iii) fraction of HCWs going through the asymptomatic pathway. We chose reasonable prior distributions for these parameters and explored the parameter space via the posterior distributions to minimize the effect of identifiability. Note that the screening probabilities are assumed to be known in the Bayesian calibration set-up, as explained in the answer to comment 3. The reported cases are in fact different from the quarantined cases (see figure 2 A vs B) and there is a significant jump in the quarantined case on January 19, due to the hospital policy change. We have now commented on the issue in the Appendix.

Reply to reviewer #2:

This paper examines the extent of COVID-19 infections in a non-COVID-19 hospital setting. The goal is to understand how COVID-19 outbreaks among health-care workers (HCW) in these settings can be reduced by non-pharmaceutical measures. The hospital setting involves patient and HCW populations (other populations, such as visitors and staff are not included). These populations are subdivided into S (individuals susceptible to infection), E (pre-symptomatic or asymptomatic infectious individuals), I1 (first stage symptomatic infectious individuals), I2 (advanced stage symptomatic infectious individuals), and R (removed or immune infected individuals). The epidemic outbreak pathway in this setting depends on (1) daily contacts between and among patients and HCW, (2) levels of infectivity moderated by facemasks, proximity, timing of visits, and other protective measures, and (3) testing and isolation based quarantine measures.

The analysis uses an individual-based modeling (IBM) methodology. A Bayesian method was used for the identification of model parameters. The model parameters were based on calibrated data from a COVID-19 outbreak in the department of neurosurgery of Union Hospital in Wuhan, China from January 5th, 2020 to February 4th, 2020. Three intervention strategies were considered: (i) social distancing (reduced contact rates) among HCW and individual protection (facial masks); (ii) enhanced screening and isolation of infected HCW; and (iii) patient-pool control (pool size and infection level control), and individual protection via HCW personal protection equipment. The Wuhan hospital data and the model output produced a highly significant correlation. The analysis of the intervention scenarios revealed that reduction of contact rates alone had marginal impact, and the wearing of facemasks had much higher

impact for reducing the number of infectious cases and the number of workday losses. Also, HCW quarantine alone only prolonged the outbreak duration.

This study is a most valuable contribution to understanding the impact of COVID 19 on HCW, which is of great importance in the toll of COVID-19 on society. The method of study, using an IBM calibrated to an actual hospital data set, is novel, and possibly adaptable to other settings (such as schools, neighbors, and workplaces). This is an excellent work and should be accepted for publication with very minor corrections.

Response: We thank the reviewer for all supported comments.

Corrections:

1. On line 136, January 5th, 2019 should be changed to January 5th, 2020.

Response: Done.

2. There is some confusion in the descriptions on lines 162 -163, between the distinction of (i) and (iii).

Response: We thank the reviewer for pointing this out. Item (i) focuses on HCWs control, i.e., social distancing (reduced contact rates) and individual protection (facial masks) for HCW-HCW interaction; Item (iii) means patient-pool control (pool size and infection level), and individual HCW protection via PPE for HCW-patient interaction. We have now revised it accordingly in the manuscript.

3. Update Reference (30) to “Risk factors of SARS-CoV-2 infection in healthcare workers: a retrospective study of a nosocomial outbreak”, Wang X, Jiang X, Huang Q, Wang H, et al., Sleep Medicine X 2 (2020) 100028.

Response: Done.

4. Update other references

Response: Done.